

# Application of hydrophilic polymers for the preparation of tadalafil solid dispersions: micromeritics properties, release and erectile dysfunction studies in male rats

Mohammed Muqtader Ahmed[1], Md Khalid Anwer[1], Gamal A. Soliman[2,3], Mohammed F. Aldawsari[1], Abdul Aleem Mohammed[4], Sultan Alshehri[5,6], Mohammed M. Ghoneim[5], Amer S. Alali[1], Abdullah Alshetaili[1], Ahmed Alalaiwe[1], Sarah I. Bukhari[6] and Ameeduzzafar Zafar[7]

[1] Pharmaceutics, Prince Sattam Bin Abdulaziz University, Alkharj, Saudi Arabia
[2] Department of Pharmacology and Toxicology, College of Pharmacy, Prince Sattam Bin Abdulaziz University, Alkharj, Saudi Arabia
[3] Department of Pharmacology, College of Veterinary Medicine, Cairo University, Giza, Egypt
[4] Department of Pharmaceutics, College of Pharmacy, Najran University, Najran, Saudi Arabia
[5] Department of Pharmaceutical Sciences, College of Pharmacy, AlMaarefa University, Ad Diriyah, Saudi Arabia
[6] Department of Pharmaceutics, College of Pharmacy, King Saud University, Riyadh, Saudi Arabia
[7] Department of Pharmaceutics, College of Pharmacy, Jouf University, Sakaka, Al-Jouf Saudi Arabia

Corresponding author
Md Khalid Anwer,
mkanwer2002@yahoo.co.in

## ABSTRACT

The objective of the present study was to improve the dissolution rate and aphrodisiac activity of tadalafil by using hydrophilic polymers. Solid dispersions were prepared by solvent evaporation-Rota evaporator using Koliphore 188, Kollidon® VA64, and Kollidon® 30 polymers in a 1:1 ratio. Prepared tadalafil-solid dispersions (SDs) evaluated for yield, drug content, micromeritics properties, physicochemical characterizations, and aphrodisiac activity assessment. The optimized SDs TK188 showed size ($2.175 \pm 0.24$ µm), percentage of content ($98.89 \pm 1.23\%$), yield ($87.27 \pm 3.13\%$), bulk density ($0.496 \pm 0.005$ g/cm3), true density ($0.646 \pm 0.003$ g/cm3), Carr's index ($23.25 \pm 0.81$), Hausner ratio ($1.303 \pm 0.003$) and angle of repose ($<25°$). FTIR spectrums revealed tadalafil doesn't chemically interact with used polymers. XRD and DSC analysis represents TK188 SDs were in the amorphous state. Drug release was $97.17 \pm 2.43\%$ for TK188, whereas it was $32.76 \pm 2.65\%$ for pure drug at the end of 2 h with 2.96-fold increase in dissolution and followed release kinetics of Korsmeyer Peppa's model. MDT and DE were noted to be 17.48 minutes and 84.53%, respectively. Furthermore, TK188 SDs showed relative improvement in the sexual behavior of the male rats. Thus the developed SDs TK188 could be potential tadalafil carriers for the treatment of erectile dysfunction.

## INTRODUCTION

In the recent past, novel drug development approaches were implemented to improve the therapeutic efficacy of new drug application (NDA), new molecular entity (NME), and new chemical entity (NCE) entities unveiling solubility and permeability issues. Approximately 75% of the newly developed chemical entities and 40% of marketed drug products reflect on being poorly water-soluble compounds (*Wlodarski et al., 2015*). The complexity associated with the pharmaceutical development for these chemical entities has majorly shifted the attention of formulation scientists, which led to various formulation approaches to enhance the solubility and dissolution profile of these entities. Phosphodiesterase-5 (PDE5) Inhibitors drug molecules are used to treat pulmonary hypertension and erectile dysfunction that exhibit poor aqueous solubility, low dissolution rate, which presents major challenges to develop into formulation. These problems could be resolved by increasing the solubility, enhancing the dissolution rate of various means such as nano/micro-particle based formulations, amorphous SDs, lipid-based drug delivery systems, pro-drugs, and co-crystals to improve the bioavailability of drugs. One of the most prevalent sexual disorders in men is erectile dysfunction (ED), affecting around 8% of men in the ages of 40 and almost 40% of men in their age of 60–69 years and found predominant in patients with diabetes mellitus (*Shamloul & Ghanem, 2013*; *Fahmy & Aljaeid, 2018*). ED has a significant impact on quality of life causes depression, anxiety, and loss of self-confidence. Moreover, ED is considerably associated with increased risk of cardiovascular diseases, stroke, and all-cause mortality independent of conventional cardiovascular risk factors (*Dong, Zhang & Qin, 2011*).

Tadalafil (TFL) is a white crystalline powder with an empirical formula ($C_{22}H_{19}N_3O_4$) and a molecular mass of 489.40 g mol$^{-1}$ (*Aboul-Enein & Ali, 2005*; *Reddy, Reddy & Reddy, 2010*). Chemically, TFL is (6R, 12aR)-6-(l,3-Benzodioxol-5-yl)-2,3,6,7,12,12a-hexahydro-2-methyl-pyrazinol [1′, 2′: l,6] pyrido [3,4- b] indole-l,4-dione (*Cialis, 2021*). TFL is a selective cyclic guanosine monophosphate (cGMP) specific phosphodiesterase 5 (PDE5) inhibitor used in the treatment of first-line erectile dysfunction (ED), benign prostatic hyperplasia (BPH) and pulmonary hypertension (*Choi et al., 2019*; *Yang et al., 2019*). In ED, TFL acts by inhibiting PDE5, stimulating the synthesis of cGMP causes the release of nitric oxide leading to corpus cavernosal smooth muscle relaxation and increased blood flow into the corpus cavernosum during sexual stimulation leading to penile erection (*Wright, 2006*). TFL, in contrast to other PDE5 inhibitors, exhibits prolonged action (about 36 h) with lessened vision irregularities (*Porst et al., 2003*; *Doggrell, 2005*).

TFL is a US-Food and Drug Administration (FDA) approved drug (2003) firstly commercialized as Cialis®, available in a dose of 2.5 to 20 mg. An oral administration attains maximum plasma drug concentration between 30 min to 6 h, exhibiting 94% bound to plasma protein metabolized by Cytochrome P450 3A4 (CYP3A4) with a mean terminal half-life of 17.5 h excreted predominantly as methyl catechol glucuronide metabolite through feces and urine (*Cialis, 2021*). According to the Biopharmaceutical classification system (BCS) TFL has been categorized as a Class II drug exhibiting poor aqueous solubility (0.02 mg/mL) with a good permeability profile, attributes focus of many researchers towards

the development of various formulations approaches to enhance its aqueous solubility and bioavailability profile (*Sakamoto et al., 2016*; *El-Badry et al., 2014*). Among the different approaches to improve the solubility of poorly water-soluble chemical entities, the most extensively focused area covers the use of SDs technique since the first attempt reported by *Sekiguchi & Obi (1961)*. SDs term was defined by Chiou and Riegelman in 1971 as dispersion of hydrophobic drug in a hydrophilic carrier prepared by fusion or solvent evaporation method (*Chiou & Riegelman, 1971*). The various possible potential approaches focused by formulation scientists to enhance the aqueous solubility of poorly water-soluble drug TFL were SDs (*Choi & Park, 2017*; *Choi et al., 2017b*), nanostructured lipid carriers (NLCs) (*Baek et al., 2015*), and complexation with cyclodextrins (*Badr-Eldin, Elkheshen & Ghorab, 2008*), self-emulsifying drug delivery systems (*El-Badry et al., 2014*), and incorporation in microporous silica (*Mehanna, Motawaa & Samaha, 2010b*), nanoparticles (*Obeidat & Sallam, 2014*) and nanocrystals (*Bhokare, Kendre & Pande, 2015*). Besides these, the SDs approach draws the attention of formulation scientists due to ease of processing methods and devices, composition flexibility, high effectiveness, low batch variability, and a wide choice of carriers. Moreover, the development of SDs of hydrophobic drugs provides promising results enhancing dissolution and bioavailability profile by attaining improved wettability, reduced particle size, increased surface area, and reduced crystallinity (*Huang & Dai, 2014*). TFL poor aqueous solubility, non-ionizable hydrophobic nature at all physiological pH with Pka value of 16.68 makes it unsuitable to develop salt form or ionization to improve its solubility and dissolution rate and high melting point of about 300 °C higher than the processing temperature of the polymers utilized to develop solid dispersion marks it as a suitable choice to develop SDs. The various approaches used to prepare SDs of TFL include hot melt extrusion (*Krupa et al., 2017*), solvent evaporation (*Choi et al., 2017a*), melting method (*Mehanna, Motawaa & Samaha, 2010a*), ball milling (*Nowak et al., 2019*), freeze drying (*Wlodarski et al., 2015*), supercritical anti-solvent (SAS) technique (*Park et al., 2014*) and spray drying (*Wlodarski et al., 2014*).

In previous studies, the polymers used to develop solid dispersions of tadalafil were polyvinylpyrrolidone (PVP) (*Al-Shdefat et al., 2019*), methylcellulose (MC), Hydroxypropyl methylcellulose (HPMC), PVP-vinyl acetate (PVP-VA) copolymer, Kollicoat IR and Soluplus® (*Wlodarski et al., 2015*), tartaric acid and Soluplus® (*Aldawsari et al., 2021*), tartaric acid and PVP/VA S-630 (*Choi et al., 2017b*), malic acid and PVP/VA S-630 (*Choi & Park, 2017*) and Kollidon®12 PF, Kollidon® VA 64, and Soluplus® (*Slámová et al., 2020*) were used to prepare SDs of TFL. The use of Kollidon®12 PF, Kollidon® VA 64, and Soluplus® to develop SDs of TFL using solvent evaporation, hot-melt extrusion, and spray drying revealed an enhanced dissolution profile exhibited by a formulation containing both Kollidons and the drug release was retarded due to the presence of soluplus® due to increased swelling of hydrophilic polymers with an increase in molecular weight during the dissolution (*Slámová et al., 2020*). Similarly, improved dissolution characteristics of TFL-SDs were achieved by using poloxamer 407 as a hydrophilic carrier using the hot melting method (*Vyas et al., 2009*). Moreover, TFL-SDs of submicron size were developed by using Kollidon PVP 30 by supercritical antisolvent (SAS) method resulted in an enhanced dissolution rate shown by 200 nm-sized solid dispersion (*Shamma*

& *Basha, 2013*). Our previous investigation aimed to develop amorphous solid dispersion of TFL by spray drying technique using natural carrier glycyrrhizin. The results showed the amorphous state of TFL in the drug-carrier binary mixture with reduced particle size and enhanced dissolution profile (*Ahmed et al., 2020b*).

Hydrophilic polymers were selected based on the previous reports and their affinity to change the crystalline drug to an amorphous form in SDs system. Among the various PVP-based polymers, Kollidon® 30 (PVP 30K grade) was found to exhibit optimum balance between polymer grade and dissolution rate based on molecular weight, which intern affects the dissolution behavior. The mechanism contributing to improved solubility by PVP-based polymers is attributed to the release of molecularly dispersed amorphous drugs and particle size reduction. Copovidone (Kollidon® VA64; Sigma Aldrich, St. Louis, MO, USA) polymer-based SDs attains physical stability attributed to intermolecular interactions, and amorphic of the poorly water-soluble drug is the major dissolution mechanism (*Nair et al., 2020*). Among the various surfactants, Kolliphor® P188 is one of the most commonly and significantly used carriers for the formulation of solid dispersions (*Ali, Williams & Rawlinson, 2010*). The addition of surfactant or self-emulsifier significantly improves the solubilization of drug as well as the release rate and overcomes the problem of precipitation and recrystallization of the drug in SDs system.

The present investigation aims to develop TFL-SDs by solvent evaporation - rota evaporator using hydrophilic polymers Kolliphor® P188, Kollidon® VA64, and Kollidon® 30 polymers. The prepared solid distributions were evaluated for physicochemical, micromeritics, drug-polymer compatibility characterization, and dissolution parameters. Furthermore, the selected TFL-polymer dispersions were evaluated for potential aphrodisiac activity by sexual behavior studies in male rats.

## MATERIALS AND METHODS

Tadalafil drug was obtained as a gift sample from Riyadh Pharmaceuticals-Riyadh, Saudi Arabia. Hydrophilic polymers; Kollidon-30 (Povidone) lot #60633424U0 and Kollidon VA-64 (Copolyvidone) lot# 4330897V0 were procured as a sample without commercial value from BASF SE, Ludwigshafen, Germany. Kolliphor® P 188 (Poloxamer 188) lot #BCBL1574V and Ethanol (99.8%) lot# SZBE3420V were purchased from Sigma Aldrich Saint Louis, USA. Milli-Q water processed in Milli-Q® Direct 8 water purification system was used throughout the studies. All the other chemicals were of analytical grade and used without further purification.

All animals used during aphrodisiac activity were procured from the animal care unit at the "College of Pharmacy, Prince Sattam Bin Abdulaziz University, Alkharj, Saudi Arabia". The study protocol was approved by the "Bio-Ethical Research Committee (BERC) at Prince Sattam Bin Abdulaziz University under reference number BERC 005-05-19. All animals were alive at the end of the study and were further utilized in other preliminary experiments.

**Table 1** Composition of prepared TFL-SDs.

| SDs code | TFL (mg) | Polymers (mg) | | | Drug: Polymer (w/w) |
| --- | --- | --- | --- | --- | --- |
| | | Kolliphor® P188 | Kollidon®64 | Kollidon®30 | |
| TK188 | 500 | 500 | – | – | 1:1 |
| TK64 | 500 | – | 500 | | 1:1 |
| TK30 | 500 | – | – | 500 | 1:1 |

## Preparation of TFL- SDs by solvent evaporation

Three batches of drug-polymers SDs in the weight ratio (1:1w/w) were prepared by dissolving TFL and hydrophilic polymers; Kolliphor® P188, Kollidon®-VA64 and Kollidon® 30, separately in 50 mL of 1:1v/v water and ethanol solvent mixture. The hydroalcoholic solution containing the drug and polymer was then placed in ultrasonication for 5 min. Thereafter, it was transferred to a round bottom flask, and the solvent was evaporated for 5 h on rota-evaporator, Buchi Rotavapor R-215 at 60 °C with 50 rpm flask rotation. The collected SDs were dried under vacuum to remove the residual solvent and pulverized. The dried product was packed in vials and labeled as TFL: Kolliphor® P188 (TK188), TFL: Kollidon®-VA64 (TK64), and TFL: Kollidon® 30 (TK30), respectively (Table 1). SD batches were stored at room temperature (20 ± 2 °C) until further testing and analysis (*Mesallati, Umerska & Tajber, 2019*).

## Practical yield percentage calculations

Practical yield (%) results reflect the efficiency of the production method and technology parameters. It's an essential tool for evaluating the cost-effectiveness of the batch. Practical yield (%) can be calculated by using the weight ratio of the precursors and product as indicated in the below equation.

$$\text{Practical yield}(\%) = \frac{\text{Weight of the precusors}}{\text{Weight of product}} \times 100.$$

## Drug content estimation

SD equivalent to 5 mg of TFL was dissolved in ethanol (10 mL), vortexed for one minute. The solution is then filtered through a 0.2 μm syringe filter (Millex$^{TM}$-LG, Hydrophilic PTFE) and diluted suitably. TFL content was estimated by UV-spectroscopy at $\lambda_{max}$ 285 nm using UV-Visible Spectrophotometer. (Jasco, V630). All the samples were analyzed in triplicate to get the standard deviation (*Mesallati, Umerska & Tajber, 2019*).

$$\text{Drug content}(\%) = \frac{\text{The absorbance of the SD sample}}{\text{Absorbance of TDL}} \times 100.$$

## Micromeritic-powder characterization
### Microscopic examination
SDs powder size was examined using a microscopic technique using a monocular compound microscope fixed with 60x magnification eye-lens with tungsten light. Staged engraved slide and eyepiece scale coincides and validated, powder understudy was smeared on stage slide, and focal length was adjusted for fine image capturing and size measurement.

### Bulk and tapped density measurement
Bulk density and tapped density were determined by placing the prepared SDs into a graduated cylinder, and volume was measured. The cylinder was then tapped until the volume showed constant value (10, 500, 750, and 1,250 time tapping; changing not more than 1 mL) then finally, the tapped volume was noted. All the values were taken in triplicate (Pharma Test PT-TD200; Hainburg, Germany).

$$\text{Bulk density} = \frac{\text{weight of sample}}{\text{Bulk volume of powdered sample}}$$

$$\text{Tapped density} = \frac{\text{weight of sample}}{\text{Tapped volume of powdered sample}}.$$

### Carr's Compressibility Index and Hausner Ratio
Flowability of bulk powders can be indicated by Carr Index (CI) and the Hausner Ratio (HR). Flowable powders means an irreversible deformation of a powder by external force leading to flow. These are the indirect parameters that measure the bulk properties of powders, as suggested in USP-Chapter 1174. The CI and HR values presented by scientists Ralph J. Carr, Jr. and Henry H. Hausner; these results could be correlated to the compressibility and flowability of powdered samples or granular material, respectively. CI is indirectly related to the particle size, flow rate, and cohesiveness, whereas HR is related to interparticle friction.

$$\text{CI}(\%) = 100 \left( 1 - \frac{\text{Bulk density}}{\text{Tapped density}} \right)$$

$$\text{Hausner Ratio} = \frac{\text{Bulk volume}}{\text{Tapped volume}}.$$

### Angle of repose
The angle of repose (AR) represents the cohesion (intraparticle) force of the particle. It's an angle of the heaped cone of the powder on the flat surface. This study was conducted in a Pharma test (PTG-S4 Automated Powder Flow Analyzer; Hainburg Germany). The sample(s) under investigation was placed in a cone; once the test starts, the aperture opens,

and powder starts flowing. The two infra-red sensors measured the rate of flow and height of the cone to the tip; the cone angle can be calculated and printed.

$$\tan\theta = \frac{\text{Height of the powder cone tip}}{\text{Radius of the powder surface}}.$$

## Fourier Transform Infrared (FTIR) Spectroscopy

FTIR spectrums of polymers; K30, K64, K P 188, TFL, and their respective SDs were acquired by triturating them in KBr individually and making the thin-translucent pellet after compression. A Prepared KBr-sample pellet was placed into the pre-validated FTIR spectrometer (FT/IR-4000; Jasco, Japan). Infrared spectra from the interferometer were obtained by processing the sample signals. IR radiation of wavenumber region from 400–4,000 cm$^{-1}$ was allowed to pass through the sample analyte, and the resulting signals were recorded, collaged, and interpreted for the possible polymer-drug interactions.

## Differential scanning calorimetric analysis

Pure drug (TLF) and prepared three SDs were sealed into the hemispherical aluminum pan by compression. Then the sample (5 mg) loaded pan was placed inside the DSC sample unit, the empty pan was also placed beside it, which acted as a reference. The samples were exposed to controlled temperature from 25–305 °C scan rate 20 °C/min, the differential scanning calorimeter (DSC N-650, SCINCO, Seoul, Korea) was connected with nitrogen supply (10 mL min$^{-1}$) (*Khuroo et al., 2018*). Thermal analysis was performed to confirm the crystalline state of the TLF inside hydrophilic polymers' SDs.

## X-ray Powder Diffraction (XRD) analysis

Diffraction pattern of polymers (K30, K64, and K P 188), pure drug TFL and SDs were obtained using diffractometer Ultima-IV (Tokyo, Japan) equipped with Ni-filtered Cu-K$\alpha$ as an ionized radiation X-ray source. Instrument was run using a Cu tube (Anode) at a generator of 40 kV using current 40 mA, 2-theta peak patterns were collected for each sample with scanned rate of 0.02o /min. X-ray beam reflections from the samples are monitored in Counts/sec as a function of the scattered angle.

## Scanning electron microscopy

The SEM Zeiss EVO LS10 (Cambridge, United Kingdom) was operated under vacuum. The samples under study (K30, K64 and K P 188) and their respective SD were mounted on the stubs held by two-sided adhesive carbon tape. Current conduct was achieved by a gold (Au) coat on the mounted sample at 20 mA using argon gas atmosphere. The beam of electron passed through the sample detected, converted to voltage, and amplified. The images were captured by tuning the desired zone at 15 kV current voltage (*Khuroo et al., 2014*).

## *In-vitro* dissolution studies

Prepared SDs were studied for *in-vitro* dissolution study to interpret the mass conversion rate of TFL from solid to the solution. The sample (SDs equivalent to 50 mg of TFL)

was dispersed into 900 mL of 0.1 N HCl dissolution medium (DM) containing 0.25% of dodecyl sulfate sodium salt. USP-II paddle dissolution apparatus (Erweka DT 600, Heusnstamm, Germany) was used under controlled conditions of $37 \pm 0.5\,°C$ temperature and rotation speed 50 rpm. The run time was 90 min; at pre-determined time intervals, five mL samples were withdrawn and replaced with fresh DM. Then pre-filtered samples were suitably diluted and analyzed spectrophotometrically at $\lambda_{max}$ 285 nm using UV-Visible Spectrophotometer (V630; Jasco, Japan). The mean of at least three aliquots readings was used to plot the drug dissolution-time profiles (*Mesallati, Umerska & Tajber, 2019*).

### Drug release kinetic models and dissolution parameters

To analyze the mechanism of the drug, release the dissolution was within model-dependent kinetic equations. The percentage drug release values were plotted against time in zero-order kinetics. Log percentage drug release was plotted against the time (min) for first-order kinetics. The percentage of drug release was plotted against the square root time for the Higuchi release model. The cubic root of the initial drug concentration was subtracted by the cube root of the percentage drug remaining against the time (min) profile used to calculate the Hixson-Crowell model. The log value of percentage drug dissolved is plotted against log time for the Korsmeyer-Peppas model.

Zero order release kinetics model: $FD = \mathrm{kxt}$
First order release kinetics model: $\int FD = \mathrm{kxt}$
Higuchi's release kinetics model: $FD = \mathrm{k}\sqrt{\mathrm{t}}$
Hixson-Crowell model: $1 - (1 - FD)^{1/3}\mathrm{K}_{1/3}\mathrm{t}$
Korsmeyer-Peppas release kinetics model: $FD = \mathrm{ktn}$

where, FD stands for a fraction of drug release, k is the release rate constant, t is time-intervals (min), and n is the release exponent of the Korsmeyer-Peppas model. Exponent n value determined the release mechanism; if $n \leq 0.45$, then the drug release follows the Fickian diffusion (case I diffusional), $0.45 < n < 0.89$ referred to anomalous (non-Fickian) diffusion. Model-independent dissolution parameters such as mean dissolution time and dissolution efficiency were calculated for the pure drug and all three batches of SD. Mean dissolution time (MDT) is determined from the area of the dissolution curve as a function of time. MDT values describe the polymer's drug retardations property and release rate; higher values indicate retarded release. MDT value could be ordered as a function of solubility and dissolution rate, which could be calculated from the following equation.

$$\mathrm{MDT} = \frac{\sum_{j=1}^{n} \mathrm{t}^{\wedge}j\,\Delta\mathrm{Mj}}{\sum_{j=1}^{n} \Delta\mathrm{Mj}}$$

where j is the number of samples, n is the number of dissolution sample times, $\mathrm{t}^{\wedge}j$ is the time at the midpoint between tj and tj $-1$, calculated by (tj+tj $-1$)/2), $\Delta$Mj is the additional amount of drug released between tj and tj $-1$.

Dissolution efficiency is defined as the area under a dissolution curve between defined time points and expressed as a percentage of the area of dissolution in the same time

described by 100% and could be calculated by the below equation.

$$\text{Dissolution efficiency}(\%) = \frac{\int_0^t y x \, dt}{y_{100} \, xt} X100$$

where y is percentage of drug dissolved at time t, y100 is the 100% drug release at time t.

### *In-vivo* aphrodisiac study in male-rats
### *Experimental design: animal- grouping and sample administration*

The study was performed as per the procedure reported by *Ahmed et al. (2020a)* healthy animals in equal sex ratio were selected and kept four rats separately into one plexiglass cage placed in a well-ventilated animal room conditioned to 22 °C $\pm$ 2 °C with humidity of 60%. All the cages have access to food pellets and water ad libitum. Animal-laden cages were housed in a reversed dark cycle (light from 6 p.m to 6 a.m) for four weeks before the study was scheduled. The animal was screened based on the sexual behavior for male rats must show intromission within 30 s of female-mating, ejaculation within 15 min of the first intromission. Female rats were made sexually receptive, lordosis responding, and proceptivity by injecting single subcutaneous injections of estradiol benzoate (10 $\mu$g) and progesterone (1 mg) prior to pairing. A randomized design was used in the study in which three groups consisting of six rats per group were considered. Group, I was the negative control (NC), administered orally with sodium carboxymethyl cellulose (1% w/v). Group II, III was administered orally with 5 mg/kg pure TFL (STD- TFL) and optimized SDs (TEST-TK188) weight equivalent with 5 mg/kg of TFL, respectively. All the laboratory setup and experiments were performed in stillness and dark, dim light using a red fluorescent lamp.

### *Sexual behavior parameters- measurement*

Half an hour after sample(s) administration, sexually energetic male and female rats were introduced into mating cages in a 1:1 ratio under the same lighting and laboratory conditions. The sexual behavior of a male with a female was observed and parameters were monitored and analyzed.

Mount latency (ML): Time (sec) from the introduction of the receptive females until the first mount by the male with pelvic thrusting. Mount frequency (MF): Number of the mount in the stipulated time. Intromission Latency (IL): the time taken from introducing the female to the first mount and intromissions. Intromission Frequency (IF): Number of intromissions, vaginal penetration by male during copulating. Ejaculation Latency (EL): It's the time from the first intromission to ejaculation in series 1 and 2. Post-Ejaculatory Interval (PEI): it's time period between an ejaculation to the next intromission. Some additional male sexual behavior parameters measurements include.

$$\text{Copulatory efficiency }(\%) = \frac{\text{number of intromissions}}{\text{number of mounts}} \times 100$$

$$\text{Intercopulatory efficiency }(\%) = \frac{\text{number of intromissions}}{\text{number of mounts} + \text{number of intromissions}} \times 100.$$

## Stability study

The stability study was performed by packing the sample under study into a tightly closed screw-lid glass (Glass laboratory 50 ml) vial. The study was conducted as suggested by MM. et al. by exposing the package for 40 °C or 40 °C/75 ± 0.5% relative humidity (RH) for four weeks, and 60 °C for two weeks respectively in a stability chamber. Thereafter, a drug content estimation and dissolution study was performed in the identical conditions of dissolution study and drug analysis (*Ahmed et al., 2021*). The dissolution profiles of the SD (TK188) before and after the stability study were compared and the similarity index was calculated by using the equation suggested by Moore and Flanner.

$$f_2 = 50 \times log_{10} \left[ \frac{100}{\sqrt{1 + \frac{\sum_{t=1}^{n}(R_t - T_t)^2}{n}}} \right]$$

where; $R_t$ and $T_t$ are the mean dissolution value for the reference and test product, respectively at time $t$, and $n$ is the number of time points.

The $f2$ value is greater than 50 or (50–100) indicates the release profiles are similar, same or equivalent. A higher f2 value indicates similarity between the test and reference dissolution profiles.

## Statistical analysis

Statistical analysis was performed by one-way analysis of variance (ANOVA) and followed by Tukey test post hoc analysis. Results were expressed as a means of three replicates ± standard deviation; data are presented as mean ± standard error of the mean. The difference between two values considered as significant levels if $P < 0.05$. All the calculation and statistical analysis was performed by using Microsoft Office Excel 2016.

# RESULTS AND DISCUSSION

## Characterization of SDs

It has been observed that the physicochemical properties of SDs chiefly dependent on hydrophilic polymers Kolliphor® P188, Kollidon® VA64, and Kollidon® 30 used in the study.

## Yield, drug content percentage, and micromeritics-powder characterizations

The percentage yield of prepared SDs was found to be in the range of (80.81 ± 2.21– 87.27 ± 3.13%w/w). The loss of product could be due to the several unit operations involved during the preparations and sticking of SDs to the wall of equipment- glassware. The results were in agreement with *Ahmed et al. (2020a)* reported (>85%) practical yield of prepared spray-dried sildenafil amorphous SDs. Percentage yield could be improved with batch size and process optimization. The drug content of SDs was found to be between (96.17 ± 2.23 to 99.31 ± 1.00%). All three batches of SDs showed compendia limits of drug content, indicating uniform drug distribution in the polymeric dispersions. Higher drug content represents the presence of hydrophilic functional groups in the polymers. Macroscopic images are reflected in Fig. 1; particles size was measured, and the mean
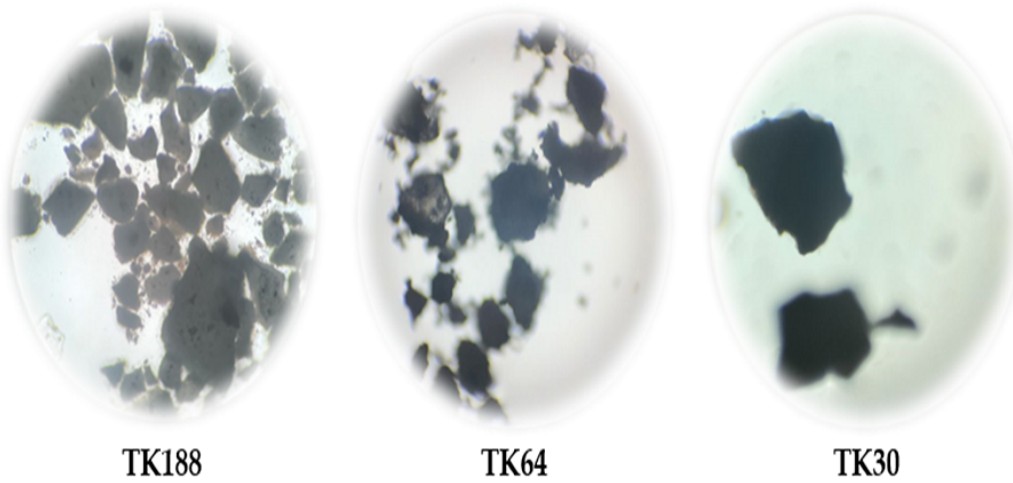

TK188        TK64        TK30

**Figure 1**   **Optical microscopic images of of TFL and its solid dispersions, TK188, TK64, and TK30.**

**Table 2**   **Physicochemical and micromeritics properties of solid dispersions.**

| SDs Batch Code | Practical yield (%) | Drug content (%w/w) | Particle size (μm) | Bulk density (g/cm3) | Tapped density (g/cm3) | Carr's index (CI) | Hausner ratio (HR) | Angle of repose ($\theta$) |
|---|---|---|---|---|---|---|---|---|
| TK188 | 87.27 ± 3.13 | 98.89 ± 1.23 | 2.175 ± 0.24 | 0.496 ± 0.005 | 0.646 ± 0.003 | 23.25 ± 0.81 | 1.303 ± 0.003 | <25 |
| TK64 | 80.81 ± 2.21 | 96.17 ± 2.23 | 4.934 ± 0.87 | 0.533 ± 0.003 | 0.600 ± 0.004 | 11.11 ± 0.11 | 1.125 ± 0.001 | <25 |
| TK30 | 84.35 ± 2.60 | 99.31 ± 1.00 | 4.448 ± 0.11 | 0.544 ± 0.002 | 0.618 ± 0.002 | 12.00 ± 0.23 | 1.136 ± 0.002 | <25 |

**Notes.**
values represent the mean ± SD of $n = 3$.

values are presented in Table 2. Particle analysis revealed the size ranges (2.175 ± 0.24 to 4.448 ± 0.11 μm) lower the particle size attributes to low bulk density and flowability. Results of micromeritics properties of all three batches of TFL SDs have been enlisted in Table 2. Bulk and true densities were found to be ranging between (0.496 ± 0.005 to 0.544 ± 0.002 g/cm$^3$), (0.646 ± 0.003 to 0.618 ± 0.002 g/cm$^3$), respectively. The bulk density of the SDs perhaps increases with an increase in the molecular weight of the polymers used. The higher true density value of TK188 represents the fluffy nature of Kolliphor® P188 polymer due to inter-particulate voids. True density indicates the compactness of powders higher the value, the more compactness of the powder, which could be due to fewer inter-particle spaces. Carr's Index and Hausner ratio show the propensity of powder under shear stress. Carr's compressibility index and Hausner's ratio range was found to be (23.25 ± 0.81 to 12.00 ± 0.23), (1.303 ± 0.003 to 1.136 ± 0.002), respectively. Inference of Carr's Index and Hausner ratio governs TK188 batch has adequate flow whereas TK64 and TK30 have the good flow. The angle of repose determines the inter-particle friction or cohesiveness of the particles. The value of $\theta$ was ≤25 for all three batches indicating non-cohesive particles and excellent flow.

## Fourier Transform Infrared (FTIR) Spectroscopy

FTIR spectroscopy showed the functional peaks of TFL in the fingerprint region, the peak for secondary amine (N-H str) at $(3,116.4 \text{ cm}^{-1})$, aliphatic-alkyl stretching $(2,258.23 \text{ cm}^{-1})$, amide ketone $(1,873.51 \text{ cm}^{-1})$, aromatic (C = C) at $(1,752.01 \text{ cm}^{-1})$. Moreover, $(1,586.16–1,503.24 \text{ cm}^{-1})$, $(1166.72–1,136.83 \text{ cm}^{-1})$ for ketone (C-O-C str symmetry) and aromatic ring present in the structure of the pure drug. This data is in good agreement with the pure TFL FTIR reported spectrum. The FTIR spectra of kolliphor®-188 showed peaks at $2,885 \text{ cm}^{-1}$ and $1,468 \text{ cm}^{-1}$ for O-H and C=O stretching, respectively. Kollidon 30 exhibited peaks at $2,957 \text{ cm}^{-1}$ (OH str) and $1,665 \text{ cm}^{-1}$ (CO str). KollidoneVA64 exhibited $2,954 \text{ cm}^{-1}$ (OH str) and $1673 \text{ cm}^{-1}$ (CO str), which are an agreement with reported spectra (*Poudel & Kim, 2021*; *Aminu et al., 2021*).

However, the FTIR spectrums of the binary SDs of TFL with Kolliphor® P188 (TK188)/ Kollidon® VA64 (TK64)/ Kollidon® 30 (TK30) is the sum of spectra of drug and polymer dispersions in which the prominent functional bands of the drug with reduced intensities observed. All the spectrums of TFL and SDs presented in Fig. 2. The FTIR spectra of TK188, TKVA64 and TK30, SDs indicated presence of polymer peaks in each respective SDs. This indicated that there were no interaction of TFL with polymers in the SDs.

## Differential scanning calorimetric analysis

All the three batches of SDs showed uniform distribution of TFL as a very fine microcrystalline structure within the hydrophilic polymer matrix of Kolliphor® P188/ Kollidon® VA64/ Kollidon® 30, Fig. 3. The sharp endothermic peak of TFL was observed at 307.5 °C, corresponding to the melting of TFL as reported previously. The reduced intensity of the thermal peak could be due to the complete liquefaction of the TFL crystal and reflects a more amorphous nature.

## X-ray powder diffraction (XRD) analysis

Prepared binary SDs represent the microcrystalline structure of TFL within the polymeric matrix, which was further confirmed with SEM. X-rays irradiated to the TFL generate a series of distinct peaks on 7.30°, 10.70°, 12.60°, 13.30°, 14.60°, 15.60°, 17.00°, 18.50°, 21.80°, 24.30° and 25.10° at diffraction angles at 2-theta axis indicating crystallinity nature of the pure drug. In the prepared SDs these peaks turn to broad and reduced in the intensity inferred as less crystalline (Fig. 4).

## Scanning electron microscopy

The microstructure of TFL active pharmaceutical ingredient (API) and its binary SDs was studied by photomicrographs captured by scanning electron microscope, represented in Fig. 5. SEM photomicrographs TFL showed the rod-shaped crystalline drug form, whereas SDs represent the ratio of drug-polymer homogeneous smooth irregular and homogeneous aspects. TK188 SDs signify a transformation of the crystalline drug into a microcrystalline/amorphous state, SDs with reduced crystallinity reported to show improved dissolution rate and bioactivity of the poorly water-soluble drug.

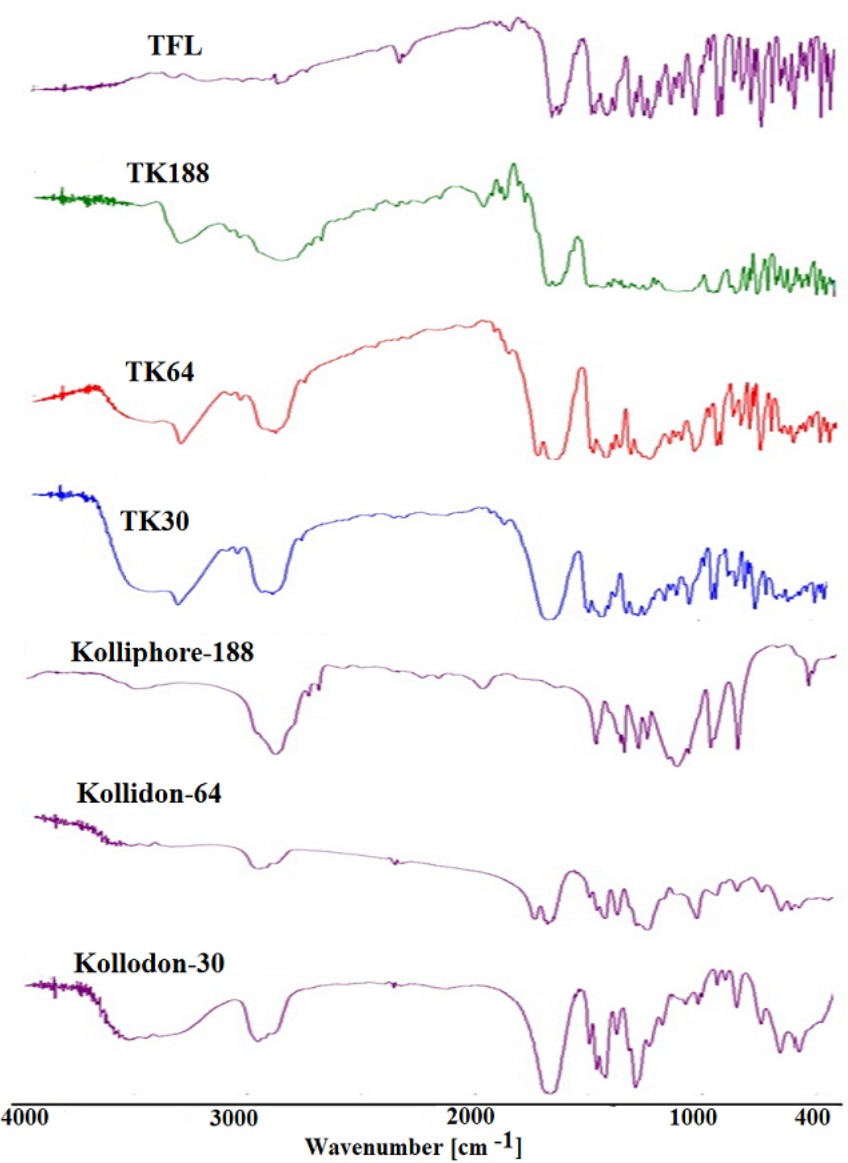

**Figure 2** FTIR spectras of tadalafil (TFL), and polymers and solid dispersions of TKP, TK64 and TK30.

### *In-vitro* dissolution, release kinetic models and dissolution parameters

Dissolution profiles of pure drugs (TFL) and three batches have been depicted in Fig. 6. The SDs of TK188, TK66, and TK30 showed $97.17 \pm 2.43\%$, $84.67 \pm 2.99\%$, and $92.67 \pm 3.23\%$ dissolution at the end of 120 min, respectively. However, the dissolution from the crystalline TFL was $32.76 \pm 2.65\%$ due to its hydrophobicity. Improved dissolution profiles were

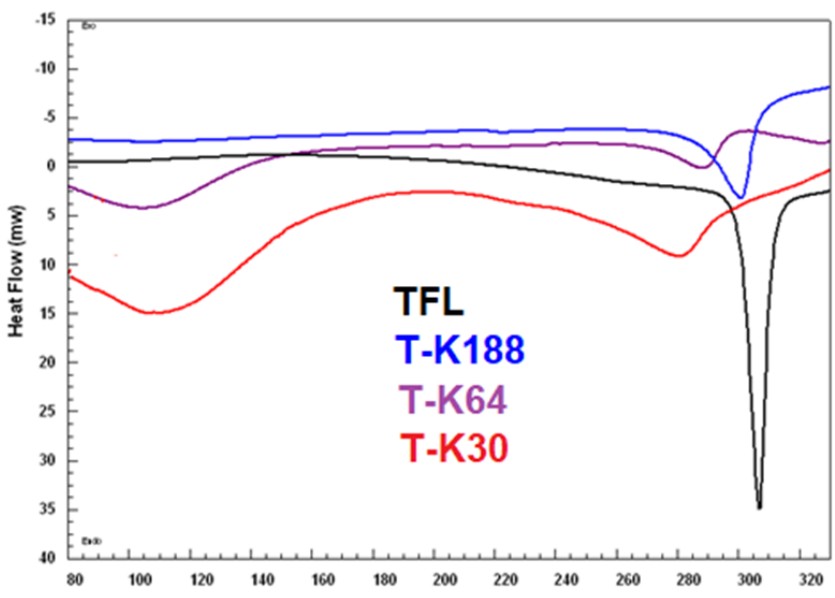

**Figure 3** DSC spectra of TFL, and their solid dispersions of TK188, TK64 and TK30.

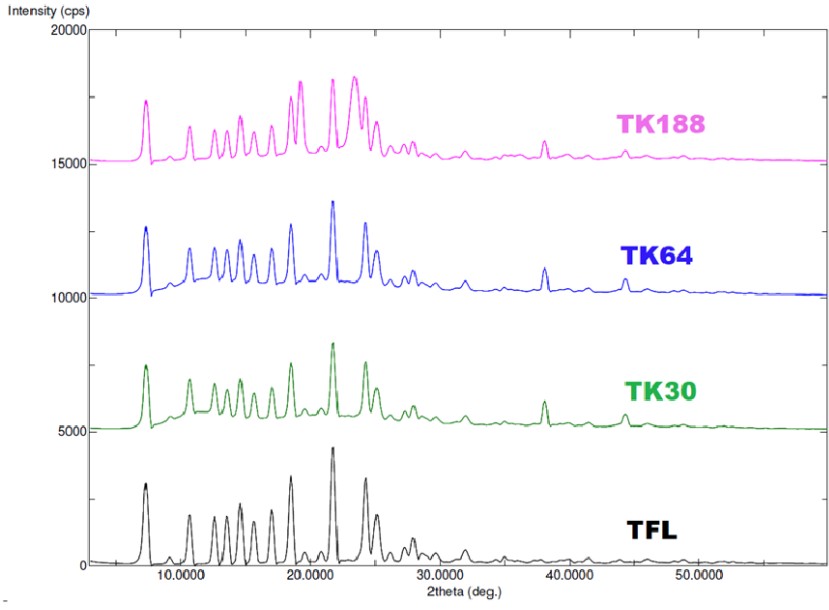

**Figure 4** XRD spectra of TFL, and their solid dispersions of TK188, TK64 and TK30.

observed for all the SD batches compared to pure TFL; the drug has changed its state from crystalline to amorphous, which was formerly confirmed by XRD and DSC studies.

The enhanced dissolution rate of SDs can be due to converting a crystalline drug into semi-crystalline to an amorphous or molecularly dispersed state, forming a higher effective surface area; more contact with the dissolution medium resulted in improved dissolution.

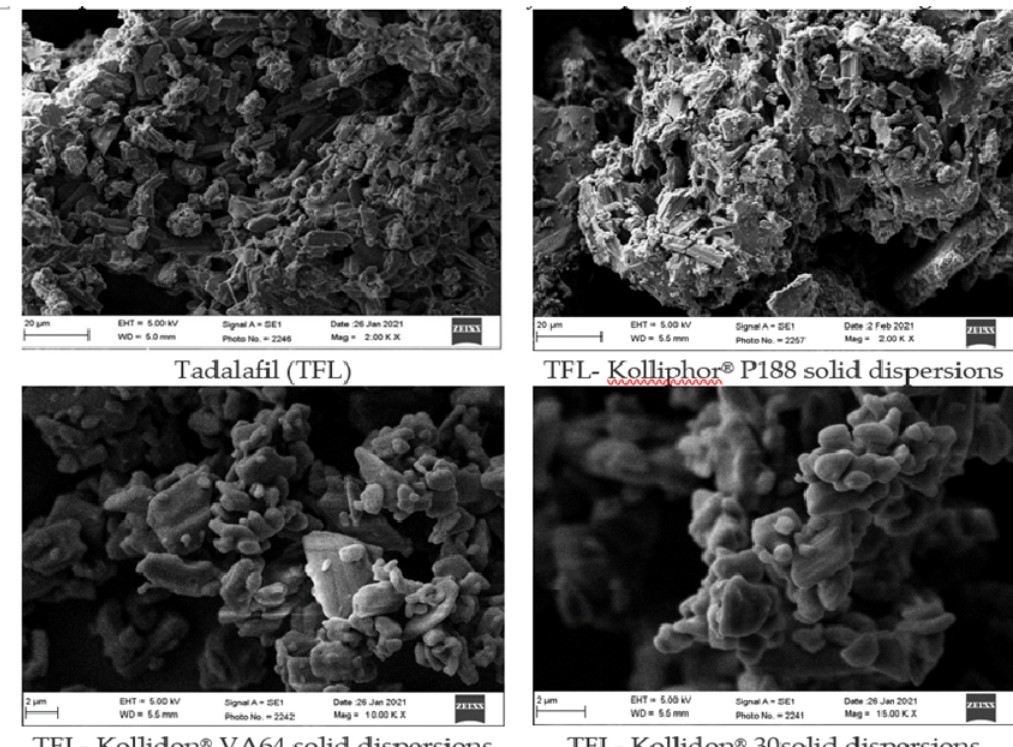

**Figure 5** SEM images of crystalline TFL, and their solid dispersions TK188, TK64 and TK30.

The relatively increased dissolution rate of Kolliphor® P188 (TK188) compared to other polymers Kollidon® VA64 and Kollidon® 30 (TK64 and TK30) most likely due to its enhanced hydrophilic and surfactant properties compared to the other agents used. The most significant improvement in dissolution rate was observed in TK188 SD, which demonstrated a 2.96 fold increase in the dissolution percentage compared to pure TFL. The highest dissolution percentage of TK188 could be due to Kolliphor® P188 wetting property, micelle formation at critical micelle concentration with hydrophilic surfaces and lipophilic core dramatically improved the solubility of hydrophobic TFL drug (*Aldawsari et al., 2021*). Model dependent kinetic equations were fitted with the dissolution data, and the regression parameters were calculated to analyze the type of drug release from binary SDs. The regression coefficient value ($R^2$) of Korsmeyer–Peppas model was found to predominate, except for TK188, for which the R2 was 0.922, indicating first-order release in which the rate dissolution is a function of the amount of the TFL remaining in the SDs. The possible mechanism of drug release proposed based on the exponent "*n*" values which were 0.328–0.369, within the range within 0–0.5, suggesting a Fickian type of drug release from the SDs. Improved drug release of TFL from the TK188 SDs was also found to correlate with the findings of release kinetic analysis. Additionally, as per the model-independent parameters, MDT was 17.48, 27.99, 23.57 min, whereas DE was 84.53%, 76.01%, 79.61% for TK188, TK60, and TK30, respectively. Whereas for pure TFL, it was 39.46 min (MDT) and 66.66% (DE). The shortest MDT was obtained for TK188 reduces the MDT by 2.25

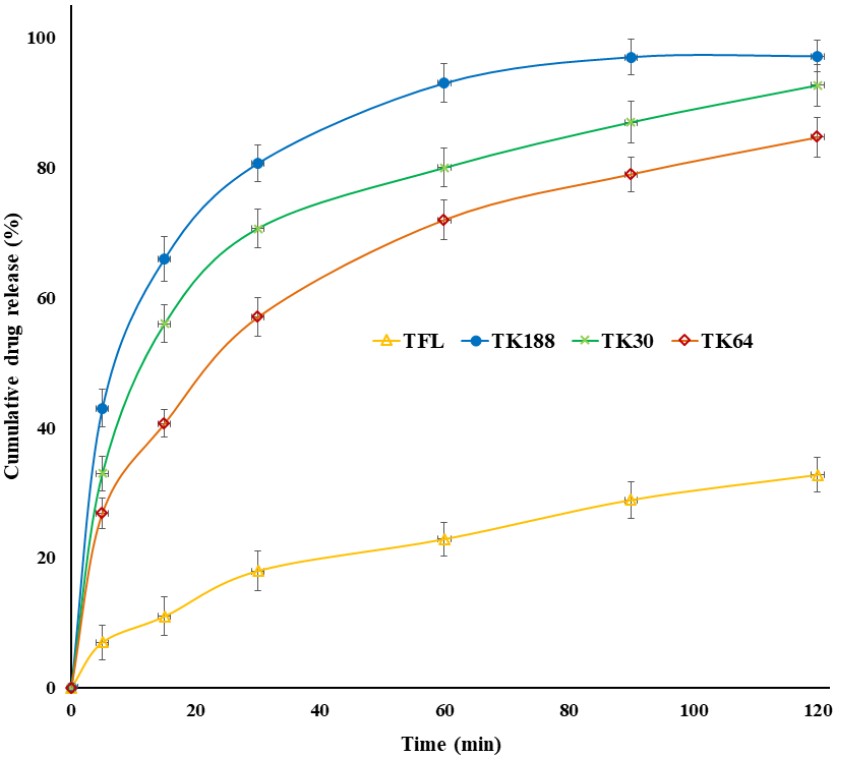

**Figure 6** *In vitro* dissolution profiles of tadalafil and their solid dispersions TK188, TK64 and TK30.

**Table 3** Drug release kinetics by model dependent mathematical processing of dissolution data and model independent dissolution parameters.

| SD batch code | Zero order | | First order | | Higuchi matrix | | Hixon Crowell | | Korsmeyer Peppas | | MDT | DE |
|---|---|---|---|---|---|---|---|---|---|---|---|---|
| | Slop | $R^2$ | Slop | $R^2$ | Slop | $R^2$ | Slop | $R^2$ | $R^2$ | n | (min) | (%) |
| TK188 | 0.717 | 0.762 | 0.013 | 0.922 | 8.569 | 0.786 | 0.027 | 0.877 | 0.219 | 0.328 | 17.48 | 85.856 |
| TK64 | 0.600 | 0.895 | 0.018 | 0.949 | 7.635 | 0.982 | 0.016 | 0.956 | 0.995 | 0.369 | 27.99 | 76.013 |
| TK30 | 0.611 | 0.841 | 0.017 | 0.960 | 8.034 | 0.955 | 0.020 | 0.940 | 0.978 | 0.313 | 23.57 | 79.614 |

**Notes.**
SD, Solid dispersion; R2, Régression coefficient; n, Diffusion coefficient; MDT, Mean dissolution time; DE, Dissolution efficiency.

fold. Therefore, Kolliphor® P188 enhances the dissolution efficiency (DE) by 1.26-fold for TK188 SDs compared to TFL alone (Table 3).

### *In-vivo* aphrodisiac activity: sexual behavior parameters-measurement

Results of sexual behavior parameters replicated in Fig. 7, all these parameters reflect significant enhancement in the sexual behavior of male rats administered with the TK188 SDs. Mount frequency (ML) and mount latency (MF) of the test group (TK188) was found to be (48.5 ± 1.57 s), (14.5 ± 0.85), as compared to the STD (56.3 ± 2.15 s), (11.4 ± 0.52), and negative control (127.2 ± 7.16 s), (4.7 ± 0.35) groups respectively. Intromission latency

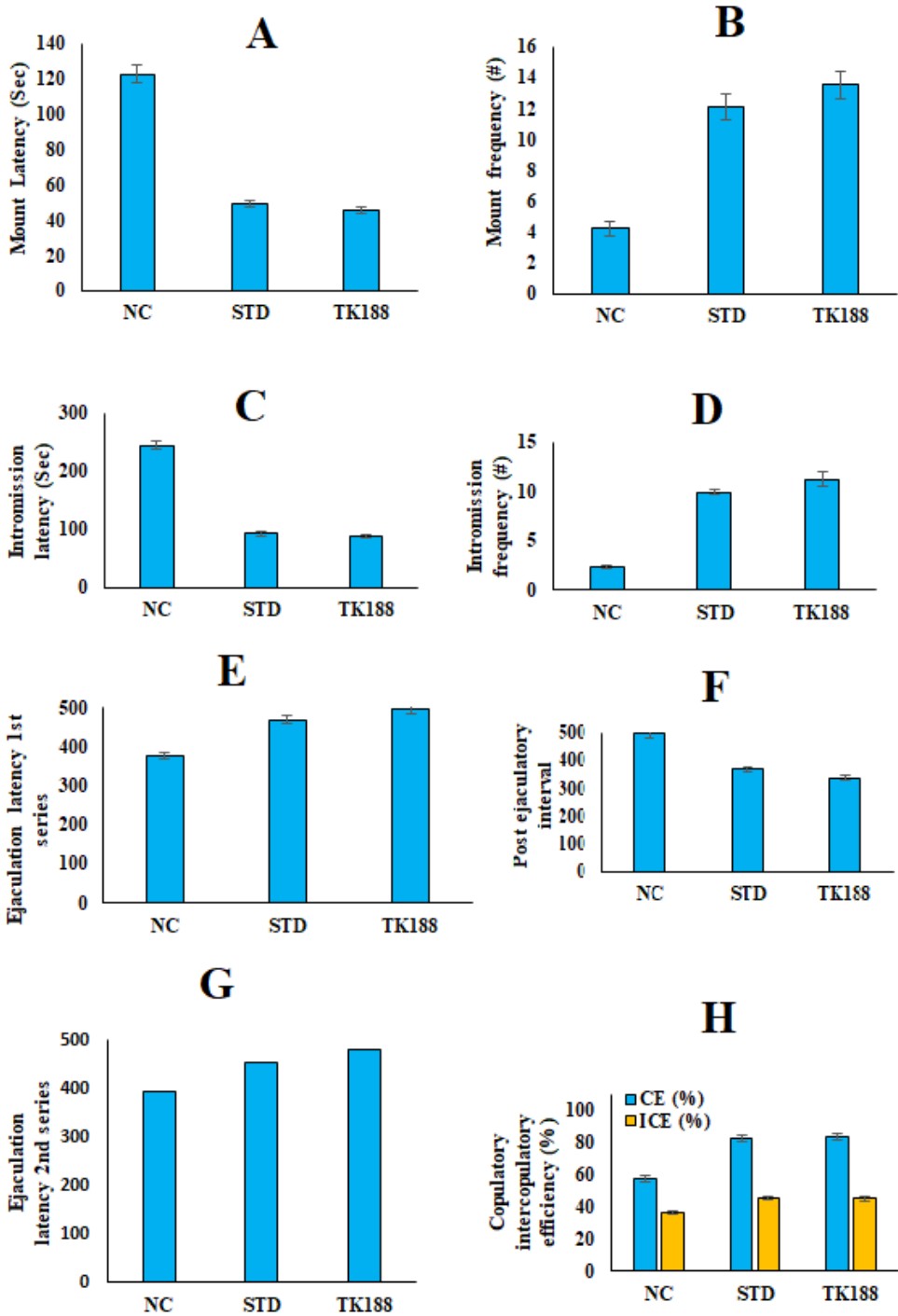

**Figure 7** *In vivo* **sexual behavior studies in rats.** (A) Mount frequency (B) mount latency, (C) intromission frequency (D) intromission latency, (E) post ejaculatory interval (F) ejaculatory latency (I Series), (G) ejaculatory latency (II Series), copulatory and intercopulat.

for the negative control, STD, and test TK188 was found (226.8 ± 9.72 s), (91.85 ± 4.75 s), and (80.60 ± 3.10 s), respectively, the time of IL was found to be significantly less for the TK188 SDs. Whereas, intromission frequency (IF) more for the tested SDs it was (12.3 ± 0.52) for TK188, and for standard it was (7.9 ± 0.27), (2.5 ± 0.17) for the negative control. The significant increases in MF and IF with corresponding decreases in ML and IL are the signs of aroused male rats, reflecting the motivation and vigor. Increased duration of EL-1 (479.4 ± 11.50 s) and EL-2 (457.6 ±10.61 s), and decreased PEI (336.7 ± 11.78 s) were observed for TK188 as compared to the STD (EL-1 (435.8 ± 12.22 s) and EL-2 (422.5 ± 11.20 s) and decreased PEI (385.8 ± 11.81 s) and negative control. The CE and IC efficiency were improved showed CE (84.83 ± 4.35%) and IC (45.90 ±1.28%) in animals exposed with TK188 in comparison to the pure TFL (STD) that showed CE (69.30 ± 3.72%) and IC (40.93 ± 1.27%), whereas in negative control CE (53.19 ± 2.15%) and IC (34.72 ± 1.47%). The results indicated the significant effect of the TK188 SDs ($P < 0.005$). Improvement in the sexual behavior of the treated (TK188) in male rats expressed potential aphrodisiac action. The improved sexual desire could be due to the stimulation of TFL to increase testosterone hormone, as reported by *Itoga et al. (2020)*. Testosterone is a steroidal hormone secreted from the testes, which assists in sexual function and desires.

## Stability study

The dissolution profiles of TK188 SDs was plotted (Fig. 8), and the data fitted into the equation, the calculated *f2* was 51.58, indicating the similarity between the cumulative percentage drug releases for the SDs (TK188) exposed for different temperature. Drug content was also estimated; only 0.9% amount of TFL was less as compared to the product content before the stability study. The observed difference in TMF content was insignificant for the TK188 SDs tested. Stability tests are carried out so that recommended storage conditions and shelf life can be included on the label to ensure that the medicine is safe and effective throughout its shelf life. The results represented that the prepared SDs (TK188) can be stable in all the climatic conditions. The data could be supportive of designating the product's shelf-life and ensuring its effectiveness during the expiration duration. Moreover, stability study data is one of the regulatory requirements for marketing approval.

## CONCLUSIONS

TFL is considered as water-insoluble with low solubility and dissolution rate. The current study could be concluded with the development of TFL-SDs improved dissolution and aphrodisiac effects can be achieved. Prepared SDs were in irregular particle shape and acceptable flow properties. The process involved renders good percentage yield, and drug content was also within the compendial limits. There was no drug-polymer physicochemical interaction and drug available as an amorphous form in SDs as per the FTIR, XRD, and DSC. Release study showed enhanced dissolution rate, MDT and DE data reflects the onset of action could be faster. Sexual parameters represent improved aphrodisiac activity with the TK188 SDs. Moreover, TK188, SDs could be stable in the different climatic zone as per

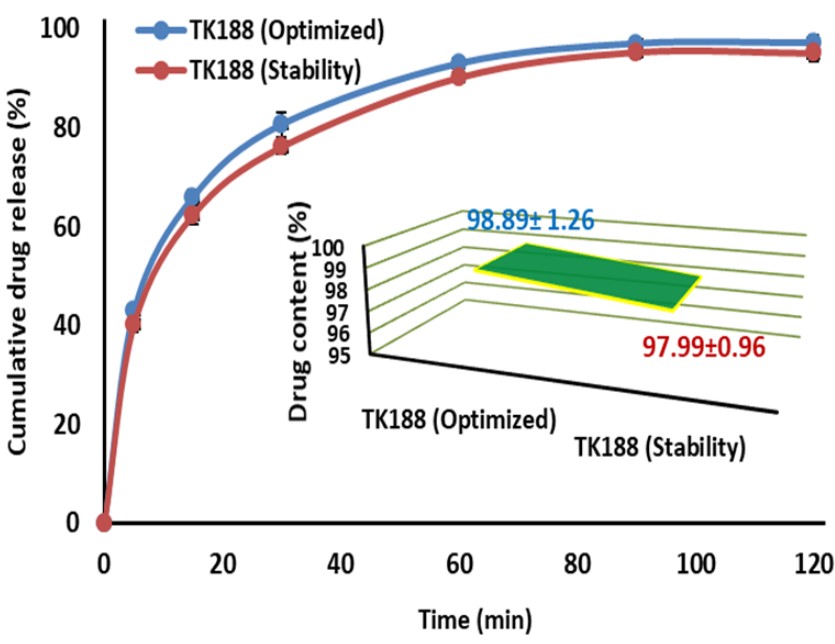

**Figure 8** Drug release, content and stability profile of optimized TK188 solid dispersion.

the stability study, without any significant deviation in the drug release profile and drug content. Therefore, TFL-Koliphore 188 (TK188) SDs could be used as a carrier matrix eliciting improved aphrodisiac activity in a short time.

### Funding
The authors were supported by the Researchers Supporting Program (TUMA-Project-2021-2), AlMaarefa University, Riyadh, Saudi Arabia. The funders had no role in study design, data collection and analysis, decision to publish, or preparation of the manuscript.

### Grant Disclosures
The following grant information was disclosed by the authors:
Researchers Supporting Program (TUMA-Project-2021-2), AlMaarefa University, Riyadh, Saudi Arabia.

### Competing Interests
The authors declare there are no competing interests.

### Author Contributions
- Mohammed Muqtader Ahmed conceived and designed the experiments, performed the experiments, prepared figures and/or tables, and approved the final draft.
- Md Khalid Anwer conceived and designed the experiments, performed the experiments, prepared figures and/or tables, and approved the final draft.

- Gamal A. Soliman conceived and designed the experiments, performed the experiments, prepared figures and/or tables, and approved the final draft.
- Mohammed F. Aldawsari conceived and designed the experiments, performed the experiments, authored or reviewed drafts of the article, and approved the final draft.
- Abdul Aleem Mohammed conceived and designed the experiments, prepared figures and/or tables, authored or reviewed drafts of the article, and approved the final draft.
- Sultan Alshehri analyzed the data, prepared figures and/or tables, and approved the final draft.
- Mohammed M. Ghoneim analyzed the data, authored or reviewed drafts of the article, and approved the final draft.
- Amer S. Alali analyzed the data, authored or reviewed drafts of the article, and approved the final draft.
- Abdullah Alshetaili analyzed the data, authored or reviewed drafts of the article, and approved the final draft.
- Ahmed Alalaiwe analyzed the data, authored or reviewed drafts of the article, and approved the final draft.
- Sarah I. Bukhari conceived and designed the experiments, prepared figures and/or tables, authored or reviewed drafts of the article, and approved the final draft.
- Ameeduzzafar Zafar performed the experiments, prepared figures and/or tables, and approved the final draft.

## Animal Ethics

The following information was supplied relating to ethical approvals (i.e., approving body and any reference numbers):

The Animal Ethics Committee, Prince Sattam Bin Abdulaziz University approved study (BERC 005-05-19).

## Data Availability

The raw data are available in the Supplementary File.

## Supplemental Information

Supplemental information for this article can be found online at http://dx.doi.org/10.7717/peerj.13482#supplemental-information.

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
