# Peer review of "Application of hydrophilic polymers for the preparation of tadalafil solid dispersions: micromeritics properties, release and erectile dysfunction studies in male rats"

_PeerJ, doi:10.7717/peerj.13482_

## Round 0.1 · original submission · Major Revisions

Please provide a comprehensively revised version addressing the editorial comments and a detailed rebuttal letter.

Reviewer 3 has suggested that you cite specific references. You are welcome to add it/them if you believe they are relevant. However, you are not required to include these citations, and if you do not include them, this will not influence my decision.

Reviewer 1 ·

Basic reporting

'no comment'

Experimental design

'no comment'

Validity of the findings

'no comment'

Additional comments

The manuscript "Application of hydrophilic polymers for the preparation of Tadalafil solid dispersions:
Micromeritics properties, release and erectile dysfunction studies in male rats (#68972)" is very well written with strong scientific pieces of evidence. I recommend acceptance for publication as is.

Reviewer 2 ·

Basic reporting

In all the manuscript, english grammar and sentences must be reviewed. For example:
Page 7, line 59. Please correct “Entity (NCE) entities...”
Page 7. Please correct sentences from line 64 to line 68.

Page 7. Indeed, choice PDE5 or PDE5I

In line 75, please write empirical formula correctly, with the numbers adequately.

Experimental design

In line 93, please indicate the tadalafil aqueous solubility.

In section 2.1. Please indicate the weight (in mg) of drug and polymer for each preparation. Insert a sample table with the preparation conditions, if necessary.

In section 2.3 Why the solution needs to be filtered? Are you sure that all the weighed drug is solubilized? Contrary, it could be generated a concentration error!

In section 2.6 Please indicate correctly the heating flow. And, it was only one scan?

In 2.10.1. The 5mg/Kg of TFL, it was also oral administered? Please specify.

Validity of the findings

In lines 115-117. Please justify the advantages of using their preparation method compared with the enlisted.

According to the described in lines 118 to 135, please justify very well the novelty of this study.

In all results and discussion sections, please indicate specifically the results related with each study. In section 3.1, please indicate the figure or table number.

The described in lines 385-387: It is not desirable the sticky SDS in the wall of equipment-glassware. In order to solve that, the glass surface must to be modified or another surfactant agent must to be added. This problem affects the stability and polydispersity of systems.

Please indicate the possible administration route of the prepared systems and justify why the 2-5 micrometer of particle size, irregularities and polydispersity could not affect.

For Figure 2, from section 3.3 must be improved. Please indicate the characteristics bands in all the FTIR spectra. Indeed, add the FTIR spectra off all polymers without TFDL.

In figure 3, endo and exo legends are too small.

In section lines 465-468, please indicates the following: The CMC (concentration micellar critical) for the Kolliphor p188. Is this CMC rebased under the preparation conditions? If micelles are formed, why the irregular morphology obtained? Please explain how the polymer and TFL could be distributed in the prepared systems.

Reviewer 3 ·

Basic reporting

All the related comments are in the additional comments box

Experimental design

All the related comments are in the additional comments box

Validity of the findings

All the related comments are in the additional comments box

Additional comments

1. Authors have not mentioned the ratio of all the polymers in the abstract (Koliphore 188, Kollidon® VA64, and Kollidon® 30 polymers in a 1:1 ratio). There are three polymers but the ratio is mentioned for only two polymers (unknown).
2. Authors needed to perform HPLC for the estimation of Tadalafil from the formulation. The results from the UV spectrometer are not authentic most of the times because of interactions.
3. How was the EE calculated? Was it a direct or indirect method? Please mention.
4. USFDA has approved Tadalafil for oral use only, how could authors compare drug efficacy when given parenterally?
5. Authors have not done optimization for the formulation to find the appropriate ration of drug and polymers.
6. Authors need to add more relevant references e.g. doi: https://dx.doi.org/10.1016/j.ijpharm.2014.07.022, https://doi.org/10.1016/j.molliq.2018.02.091, doi: https://dx.doi.org/10.1016/j.mehy.2015.03.003, DOI: https://dx.doi.org/10.3109/10717544.2015.1105323; https://doi.org/10.1016/j.molliq.2017.11.081; DOI: https://dx.doi.org/10.26717/BJSTR.2020.31.005129

---

## Round 0.2 · Major Revisions

Unfortunately, one major concern of one of the reviewers is still unanswered, and it is critical for the discussion of micelles.

"In section lines 465-468, please indicates the following: The CMC (concentration micellar critical) for the Kolliphor p188. Is this CMC rebased under the preparation conditions? If micelles are formed, why is the irregular morphology obtained? Please explain how the polymer and TFL could be distributed in the prepared systems.

Answer: Authors raised a good query, Unfortunately, we did not perform the CMC experiment. "

According to their answer, authors can not talk about micelles and justify their discussion about micelles formation, assuring the micelles' presence, if they do not know the CMC of their preparations and that CMC is exceeded during experiments.

Therefore, the presence of micelles can not be mentioned in the discussion about it.

Please address this thoroughly in your response letter.

Reviewer 3 ·

Basic reporting

Meets expectations

Experimental design

Meets expectations

Validity of the findings

Meets expectations

Additional comments

The authors have answered all the queries satisfactorily.
I recommend the manuscript to be accepted in this journal.

---

## Round 0.3 · Minor Revisions

The point about the critical micelle concentration (CMC) is important. At a minimum, this should be mentioned in the discussion. Please explain in more detail in the discussion even if the authors have not addressed this experimentally the CMC.

---

## Round 0.4 · accepted · Accept

Thanks for addressing all the revisions and corrections requested. Now your manuscript is accepted in PeerJ.